# Identification of *Trichomonas vaginalis* 5-Nitroimidazole Resistance Targets

**DOI:** 10.3390/pathogens12050692

**Published:** 2023-05-10

**Authors:** Keonte J. Graves, Colin Reily, Hemant K. Tiwari, Vinodh Srinivasasainagendra, William Evan Secor, Jan Novak, Christina A. Muzny

**Affiliations:** 1Division of Infectious Diseases, Department of Medicine, University of Alabama at Birmingham, Birmingham, AL 35233, USA; 2Division of Nephrology, Department of Medicine, University of Alabama at Birmingham, Birmingham, AL 35233, USA; creily@uab.edu; 3Department of Microbiology, University of Alabama at Birmingham, Birmingham, AL 35233, USA; 4Department of Biostatistics, School of Public Health, University of Alabama at Birmingham, Birmingham, AL 35233, USA; htiwari@uab.edu (H.K.T.); vinodh@uab.edu (V.S.); 5Division of Parasitic Diseases and Malaria, Center for Global Health, Centers for Disease Control and Prevention, Atlanta, GA 30329, USA

**Keywords:** *Trichomonas vaginalis*, 5-nitroimidazoles, resistance mechanisms, differentially expressed genes, RNA sequencing

## Abstract

*Trichomonas vaginalis* is the most common non-viral sexually transmitted infection. 5-nitroimidazoles are the only FDA-approved medications for *T. vaginalis* treatment. However, 5-nitroimidazole resistance has been increasingly recognized and may occur in up to 10% of infections. We aimed to delineate mechanisms of *T. vaginalis* resistance using transcriptome profiling of metronidazole (MTZ)-resistant and sensitive *T. vaginalis* clinical isolates. In vitro, 5-nitroimidazole susceptibility testing was performed to determine minimum lethal concentrations (MLCs) for *T. vaginalis* isolates obtained from women who had failed treatment (*n* = 4) or were successfully cured (*n* = 4). RNA sequencing, bioinformatics, and biostatistical analyses were performed to identify differentially expressed genes (DEGs) in the MTZ-resistant vs. sensitive *T. vaginalis* isolates. RNA sequencing identified 304 DEGs, 134 upregulated genes and 170 downregulated genes in the resistant isolates. Future studies with more *T. vaginalis* isolates with a broad range of MLCs are needed to determine which genes may represent the best alternative targets in drug-resistant strains.

## 1. Introduction

*Trichomonas vaginalis* is a parasitic protozoan and the causative agent of the sexually transmitted infection (STI) trichomoniasis. Trichomoniasis is the most common, non-viral, curable STI, affecting an estimated 3.7 million people in the U.S. and over 200 million people worldwide [1,2]. Those with *T. vaginalis* infection can be symptomatic or asymptomatic, with the majority being asymptomatic [3,4]. Symptomatic women can present with vaginal erythema, discharge, genital pruritus, dysuria, and/or dyspareunia [5], while symptomatic men present with non-gonococcal urethritis, prostatitis, and epididymitis [4]. Trichomoniasis is associated with increased risk of acquisition and transmission of HIV and other STIs, as well as adverse birth outcomes and other gynecologic sequela among women [6,7,8].

Currently approved drugs for the treatment of trichomoniasis are from the 5-nitroimidazole class (metronidazole [MTZ], tinidazole [TDZ], and secnidazole [SEC]) [5,9,10]. MTZ was the first 5-nitroimidazole introduced to treat *T. vaginalis*. However, treatment failures were observed soon after its introduction [11,12], suggesting that the potential for MTZ resistance is encoded in the genome of *T. vaginalis* [13]. In a study of 568 clinical *T. vaginalis* isolates collected from women at six STI clinics across the U.S., the prevalence of low-level in vitro MTZ resistance was 4.3%. However, this study was performed over a decade ago and more contemporary data are needed [14]. We recently reviewed the literature on this topic and found six additional observational studies, including a total of 679 *T. vaginalis*-infected women; 260 women (38.3%) across these six studies had resistant *T. vaginalis* isolates [15]. However, these data are also not contemporary as these studies were conducted from 1986 to 2011. Thus, the current prevalence of *T. vaginalis* resistance among clinical isolates across the U.S. is unknown, and additional studies are needed.

As noted in our recent systematic review of the literature [16], clinical resistance to 5-nitroimidazoles in *T. vaginalis* appears to be relative and not absolute; some *T. vaginalis* infections that are resistant to standard doses of MTZ can be cured by higher doses of the medication taken for longer periods of time [5]. *T. vaginalis* treatment failure is more common with standard doses of MTZ (2.2–9.6%) than TDZ (0–2%) [16]. In addition, the continued use of MTZ and TDZ could lead to cross-resistance to other 5-nitroimidazoles, including SEC [17], as they share the same mode of action for drug activation [18]. 

Two types of 5-nitroimidazole resistance have been described in the literature: anaerobic resistance (laboratory-induced, in vitro) and aerobic resistance (clinical treatment failures) which can also be laboratory-induced [19,20]. Anaerobic resistance is characterized by decreased expression of genes/proteins involved in the two main carbohydrate/energy-metabolism pathways (pyruvate-dependent and malate-dependent) and drug activation pathways of *T. vaginalis*, such as the pyruvate:ferredoxin oxidoreductase (PFOR), ferredoxin (Fdx), malic enzyme/malate dehydrogenase (ME/MDH), NADH dehydrogenase, and nitroreductase (NTR) pathways [21,22,23,24]. 5-nitroimidazole anaerobic resistance in *T. vaginalis* has also been associated with increased glucose scavenging as well as possible alternative energy pathways involving increased lactate dehydrogenase (LDH) and/or alcohol dehydrogenase (ADH) activities [24,25,26]. Aerobic resistance in *T. vaginalis* is characterized by decreased expression of genes/proteins involved in oxygen scavenging and antioxidant defense mechanisms, such as flavin reductase 1 (FR1), thioredoxin reductase (TrxR), and thioredoxin peroxidase (TrxP) in addition to superoxide dismutase (SOD) and NADH oxidase [20,25,27,28,29].

A better understanding of the mechanisms of 5-nitroimidazole resistance among *T. vaginalis* is needed to improve the detection of resistance as well as inform the development of novel treatment options. Thus, the objective of this study was to assess gene-expression patterns in MTZ-resistant (MTZ-R) vs. MTZ-sensitive (MTZ-S) *T. vaginalis* isolates. We hypothesized that MTZ-R *T. vaginalis* isolates exhibit differentially expressed genes (DEGs) associated with MTZ activation, MTZ removal, or detoxification [30,31,32].

## 2. Materials and Methods

### 2.1. T. vaginalis Isolate Selection and Growth in Culture 

Frozen *T. vaginalis* isolates were obtained from the Centers for Disease Control and Prevention (CDC) (#252, #904) under determination #CGH-LSDB-3/6/23-def6d and from two previous studies conducted at the University of Alabama at Birmingham (UAB); IRB Protocols #300007385 and #130425010, respectively. Written informed consent had been obtained in both of the UAB studies including consent for use of stored specimens for future research (#1003, #1012, #1021, #1073, #4446, #4448) [33,34,35]. CDC *T. vaginalis* isolates #252 and #904, MTZ-R and MTZ-S controls, respectively, were used for reference drug susceptibility testing (Table 1). The six clinical isolates from UAB were stored at −80 °C and included three MTZ-R isolates: 4448 [MLC 50–100 µg/mL], 1073 [MLC 200 µg/mL], and 4466 [MLC 400 µg/mL]) and three MTZ-S isolates (1003, 1012, and 1021 [MLCs < 50 µg/mL for all]) (Table 1). 

All *T. vaginalis* isolates were grown using Diamond’s Trypticase–Yeast–Maltose (TYM) media. Frozen isolates (2 mL) were thawed and added to 9 mL of warm TYM media (37 °C) contained in 15-mL polypropylene conical tubes. Cultures were put into Mitsubishi anaerobic chambers with two AnaeroPack-Anaero pouches and then placed in an incubator at 37 °C for a minimum of 3 days to reach optimal cell density (i.e., 10^6^ cells per culture). Approximately 1 mL of the cultures were then passed into new 15-mL conical tubes with fresh TYM media every other day. An antibiotic cocktail of 100× Penicillin/Streptomycin-Amphotericin B (MP Biomedicals, Solon, OH, USA) was added to each culture tube to prevent growth of bacteria and fungi. Once optimal growth conditions had been met (10^6^ cells per culture), a 5 mL aliquot of the culture was used for 5-nitroimidazole susceptibility testing while the remaining cells were pelleted using centrifugation (2200 rpm for 10 min) and washed with sterile phosphate-buffered saline (PBS) for RNA extraction.

### 2.2. 5-Nitroimidazole Susceptibility Testing

MTZ, TDZ, and SEC resistance testing of the *T. vaginalis* isolates was performed using a modified CDC susceptibility testing protocol [36,37]. Briefly, the three 5-nitroimidazoles were solubilized in dimethyl sulfoxide (DMSO) and used to prepare 2-fold serial dilutions (400 μg/mL to 0.1 μg/mL) in Diamond’s TYM media in round-bottom or flat-bottom 96-well microtiter plates. The drug concentrations were tested in triplicate; duplicate serial dilutions of equivalent final concentrations of DMSO were included to control parasite viability. Trichomonads (10^4^/well) were added to each well and the plates were incubated at 37 °C for 46–50 h under aerobic conditions. The plates were then examined using an inverted microscope at 100× magnification to evaluate cell motility. The lowest concentration of MTZ at which no motile parasites were observed was recorded as the minimal lethal concentration (MLC). Resistance to MTZ was defined as an MLC ≥ 50 μg/mL, while resistance to TDZ was defined as an MLC ≥ 6.3 μg/mL (Table 1) [33,36,37]. The resistance breakpoint for SEC has not been previously determined; however, our ongoing study suggests that a SEC MLC > 25 μg/mL correlates with resistance (unpublished data).

### 2.3. RNA Extraction for RNA-Sequencing

Total RNA was extracted from the centrifuged and PBS-washed cell pellets using Trizol reagent (Invitrogen, Carlsbad, CA, USA), according to the manufacturer’s instructions. Briefly, cell pellets were transferred into 1.5 mL microcentrifuge tubes, 1 mL of Trizol was added, and the tubes were incubated at room temperature for 5 min. Next, 200 μL of chloroform was added to the tubes, shaken for 15 s and incubated at room temperature for 10 min. Samples were then centrifuged at 12,000× *g* for 15 min. The aqueous layer was transferred to a new 1.5 mL microcentrifuge tube where 500 μL of isopropanol was added and the tubes were incubated for 10 min at room temperature followed by centrifugation for 8 min at 12,000× *g*. The supernatant was carefully removed and the RNA pellet washed using 1 mL of 75% ethanol and centrifuged for 5 min at 12,000× *g*. The ethanol was removed, and the RNA pellet was allowed to air dry for 2–3 min before 30 μL of distilled water was added to dissolve the RNA pellet, followed by incubation at 55 °C in a heating block for 15 min to enhance RNA solubilization. After the total RNA was extracted, the quality and purity of the RNA samples were assessed through the use of a NanoDrop^TM^ Lite instrument. The purity ratio and concentration of RNA for each sample was measured and recorded (Appendix A).

### 2.4. RNA-Sequencing, Bioinformatics Analyses, and Statistical Analysis

Next-generation RNA sequencing and initial bioinformatics analysis was performed by Genewiz (South Plainfield, NJ, USA). In-house bioinformatics analyses were performed using the edgeR package of Seurat R script (Appendix A) [38,39,40]. Prior to performing differential expression analysis, the raw read-counts were averaged across groups of sample replicates. The threshold for further analysis was set for genes with a −Log10(*p*-value) greater than 5 and Log2-fold change (FC) > 3 or less than −3. Genes that met these parameters were considered to be significantly dysregulated. Transcriptomic data were subsequently cross-referenced to the trichdb.org reference database for additional analysis including gene ontology (GO) enrichment and word cloud enrichment as well as pathway mapping (metabolic pathway enrichment). Next-generation sequencing raw and curated data are available at the NCBI Gene Expression Omnibus (GEO), accession number GSE227448.

## 3. Results and Discussion

### 3.1. Differential Expression of Genes in MTZ-R vs. MTZ-S T. vaginalis Isolates

In-house bioinformatics analyses were performed to characterize the transcriptomic profiles of eight *T. vaginalis* isolates (Table 1) categorized into two distinct groups (four MTZ-R and four MTZ-S *T. vaginalis* isolates) (Figure 1). This comparison identified 304 DEGs between the resistant and sensitive *T. vaginalis* isolates with *p*-values < 0.05 (Figure 1, Appendix A). There were 14 significant differentially expressed genes with −Log10(*p*-value) >5; seven were upregulated in the resistant isolates and seven were downregulated (Table 2). 

Of the seven significantly upregulated genes, six were conserved hypothetical proteins. The six genes encoding the conserved hypothetical proteins consisted of three uncharacterized proteins (TVAG_185520; TVAG_174500; TVAG_303800) and three with predicted protein domains (TVAG_003210; TVAG_064800; TVAG_191000). TVAG_003210 has been predicted to possess SANT/Myb Homeobox-like domains. These domains are DNA-binding domains conserved across various transcription factors. TVAG_064800 (galactose-binding-like domain) and TVAG_19100 (epidermal growth factor [EGF]-like domain) possess domains commonly found in proteins that bind cell-surface receptors and proteins. The additional upregulated gene encoded a Leucine-rich repeat (LRR) protein, BSPA-like surface antigen (TVAG_474560) (Table 2). LRR proteins are expressed on the surface of *T. vaginalis* and act as virulence factors [41,42], suggesting that MTZ-resistant strains may have increased virulence. These surface proteins have important functions related to the cytoadherence of *T. vaginalis* to vaginal squamous epithelial cells. 

Genes encoding ribosomal proteins, two 50S-subunit ribosomal proteins (TVAG_345450, TVAG_345440) and one 30S-subunit ribosomal protein (TVAG_474000) involved in protein synthesis were among the significantly downregulated genes in MTZ-resistant isolates (Table 2). The other four downregulated genes encoded conserved hypothetical proteins; two were uncharacterized (TVAG_054400, TVAG_108140) and two had predicted protein domains (TVAG_604680, TVAG_070260) (Table 2). TVAG_604680 was predicted to be a Shisa-like protein. Shisa-like proteins are transcription-factor-type transmembrane proteins involved in signal transduction between the endoplasmic reticulum and the cell surface. Lastly, the downregulated TVAG_070260, similar to the upregulated TVAG_064800, encoded a protein with a galactose-binding-like domain.

### 3.2. Metabolic Pathways Associated with DEGs in MTZ-R T. vaginalis Isolates

#### 3.2.1. Upregulated Genes

A metabolic pathway enrichment analysis was performed to characterize which metabolic pathways were associated with the 304 DEGs identified from RNA sequencing. 

Four unique upregulated genes were characterized as interacting in eight different metabolic processes (Table 3 and Appendix A). The most enriched pathway was for thiamine metabolism, which included two ATP-binding cassette (ABC) transporter genes (TVAG_162060 and TVAG_222600) (Figure 2 and Appendix A). The upregulation of ABC transporters in MTZ-R *T. vaginalis* isolates has been recently observed [43]. The role of ABC transporters in drug resistance mechanisms of various protozoan parasites has been previously detailed [44]; however, they have not been well described for *T. vaginalis*. ABC transporter proteins of parasitic protozoans are transmembrane proteins that aid in a wide variety of cellular processes, which includes mediating the transportation of drugs away from their intended intracellular targets [45]. This would be consistent with increased expression of these genes in MTZ-R *T. vaginalis* isolates with these isolates being more able to excrete 5-nitroimidazole drugs, prolonging their survival in the presence of 5-nitroimidazole medications. 

An iron-dependent alcohol dehydrogenase (ADH) gene (TVAG_302980) was also one of the unique genes identified and shown to be involved in 7 of the 8 enriched metabolic pathways (Table 3). The most enriched pathway for TVAG_302980 was for methane metabolism while the least enriched pathway was for linoleic acid metabolism. A prior study using comparative 2DE analysis found that the downregulated expression of a zinc-dependent ADH-1 enzyme was correlated with MTZ-R *T. vaginalis* [25]. ADH-1 has been hypothesized to be the main enzyme involved in the production of ethanol, a minor end-product of *T. vaginalis*, through the reduction of acetaldehyde, a possible byproduct of pyruvate reduction by PFOR. A more recent study employing RNA-sequencing also observed the downregulation of an ADH gene in MTZ-R isolates [43]. However, expression of an iron-dependent ADH gene in the present study was significantly upregulated in MTZ-R *T. vaginalis* isolates. This implies that different isoforms of ADH genes may perform various functions in the resistance mechanisms of MTZ-R *T. vaginalis*.

#### 3.2.2. Downregulated Genes

Fifteen unique genes involved in five different metabolic pathways were identified during the metabolic pathway enrichment analysis of 170 downregulated genes (Table 4 and Appendix A). The most enriched pathway was riboflavin metabolism (Figure 3 and Appendix A) containing three unique genes; two genes for nitroreductase (NTR)-like conserved hypothetical proteins (TVAG_036500, TVAG_205740) and one other gene encoding a conserved hypothetical protein (TVAG_072960) (Table 4). As mentioned previously, NTR is an enzyme capable of reducing (activating) 5-nitroimidazoles [46]. Single nucleotide polymorphisms (SNPs) in two NTR genes (*ntr4*, *ntr6*) are associated with MTZ resistance [31]. 

Of the 15 unique genes, 10 coded for ankyrin repeat-containing proteins: TVAG_063860, TVAG_067220, TVAG_217780, TVAG_284100, TVAG_024820, TVAG_040800, TVAG_100390, TVAG_494870, TVAG_497170, TVAG_528020 (Table 4). Ankyrin repeats are very common protein domains involved in protein–protein interactions. These domains are present in various types of proteins and have a wide diversity of functions as transcriptional initiators, cell cycle regulators, cytoskeletal proteins, ion transporters, and signal transducers. The ankyrin repeat containing proteins identified in this study are involved in Terpenoid backbone biosynthesis (TVAG_063860, TVAG_100390, TVAG_217780, TVAG_528020), toluene degradation (TVAG_063860, TVAG_217780), aminoacyl-tRNA biosynthesis (TVAG_024820, TVAG_040800, TVAG_100390, TVAG_494870, TVAG_497170), as well as fructose and mannose metabolism (TVAG_063860, TVAG_067220, TVAG_217780, TVAG_284100) (Table 4).

### 3.3. Differential Expression of Genes Encoding Resistance-Related Proteins in MTZ-R T. vaginalis

In addition to the newly identified DEGs, we also investigated the expression of genes previously described in studies of *T. vaginalis* resistance to 5-nitroimidazoles [22,25,28,31,47]. These genes encode hydrogenosomal and cytosolic proteins involved in processes, such as energy (carbohydrate) metabolism, detoxification, and oxygen scavenging (antioxidant/redox pathway) (Figure 4, Table 2). A few of those proteins/enzymes (Fdx, NTR, TrxR) have been associated with drug activation. Genes important for energy metabolism, detoxification, and oxygen scavenging were primarily downregulated in the MTZ-R *T. vaginalis* isolates in this study (Figure 4, Table 5). From the energy metabolism pathway, one Fdx gene (TVAG_292710), four ME genes, and two NADHD genes were downregulated (Table 5). Two TrxR genes from the oxygen scavenging pathway were downregulated. TrxR, in addition to reducing MTZ, can also form a covalent adduct with the reduced MTZ anion. This in turn causes a disruption of the *T. vaginalis* redox system. [48]. This could potentially lead to decreased oxygen scavenging (increased oxygen levels), which could lead to further inactivation of MTZ through futile cycling; a key feature of aerobic resistance (Figure 4). Lastly, five NTR genes were identified as downregulated in the MTZ-R *T. vaginalis* isolates. Single nucleotide polymorphisms (SNPs) found in Fdx and NTR genes that lead to truncated non-functional proteins have been linked to MTZ resistance in MTZ-R *T. vaginalis* isolates [31,47,49,50].

## 4. Conclusions

In this study, RNA sequencing identified several DEGs in MTZ-R vs. MTZ-S *T. vaginalis* isolates. There was a noticeable difference in the gene expression patterns depending on MTZ sensitivity status. As expected, the DEGs from MTZ-resistant isolates included genes involved in various metabolic pathways relevant to 5-nitroimidazole resistance such as carbohydrate/energy metabolism, drug activation, and oxygen scavenging [19,20,22,27]. However, only nitroreductase (NTR, TVAG_205740 and TVAG_036500) and alcohol dehydrogenase (ADH, TVAG_302980) genes were significantly dysregulated with a Log2FC > 3 or <−3 and a −Log10(*p*-value) > 5. A majority of the DEGs identified in this study have not been characterized, which suggests that other genes and pathways could contribute to *T. vaginalis* 5-nitroimidazole resistance.

This study had several limitations. One limitation involved the small sample size of *T. vaginalis* isolates used. A larger sample size and the inclusion of more MTZ-R *T. vaginalis* isolates would provide more rigor and statistical significance to future investigations. This study could have also benefited from a time-course analysis of gene expression, as genes and their encoded products can have multiple interactions and gene expression is not static. In addition, investigation of differential gene expression of our *T. vaginalis* isolates in the presence of sub-lethal concentrations of MTZ, TDZ, and SEC would have also been beneficial. This study did not include an analysis of the *T. vaginalis* proteome. Proteomics would provide additional beneficial information, giving a clearer picture of which genes are being translated into proteins under varying conditions. Our results should be interpreted with caution given that they are based on mRNA expression data.

An important next step in this line of research would be to select several of the genes identified in this study and perform qPCR on a larger number of MTZ-S and MTZ-R *T. vaginalis* isolates to obtain a better understanding of which genes are most important in the overall population. Additionally, we could validate those gene targets using small interfering RNA knock-down [51,52], followed by subsequent qPCR to assess the level of gene disruption. The effects of gene-specific knockdowns would also be assessed through 5-nitroimidazole susceptibility testing.

## Figures and Tables

**Figure 1 pathogens-12-00692-f001:**
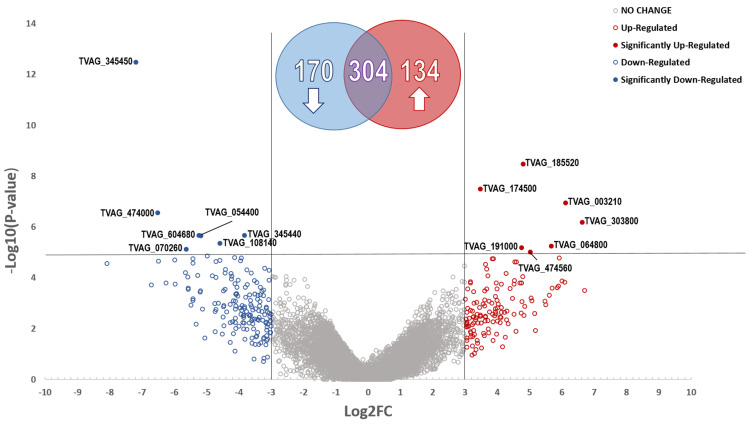
Differentially Expressed Genes of MTZR vs. MTZ-S *T. vaginalis* Isolates. Volcano plot for log2 fold change (Log2FC) plotted against the −Log10 of the *p*-value [−Log10(*p*-value)] showing the differential expression of genes between four MTZ-R and four MTZ-S *T. vaginalis* isolates. Significant DEGs with a −Log10 of the *p*-value > 5 (red- and blue-filled circles). Genes with no significant difference in expression (Grey-outlined circles). Upregulated genes with a log2 fold change greater than 3 and *p*-value < 0.05 (red-outlined circles). Downregulated genes with a log2 fold change less than −3 and *p*-value < 0.05 (blue-outlined circles). Venn diagram characterizing the expression profile of the 304 DEGs; a log2 fold change >+3 and <−3, and *p*-value < 0.05. Abbreviations: MTZ = metronidazole; DEGs = differentially expressed genes.

**Figure 2 pathogens-12-00692-f002:**
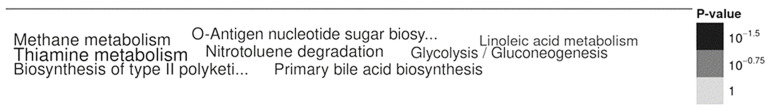
Metabolic pathway enrichment word cloud of upregulated genes. Word cloud generated using the *p*-values and full terms from the metabolic pathway enrichment analysis (Table 3) via a program called GOSummaries (trichdb.org).

**Figure 3 pathogens-12-00692-f003:**
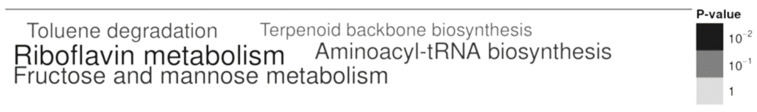
Metabolic pathway enrichment word cloud of downregulated genes. Word cloud generated using the *p*-values and full terms from the metabolic pathway enrichment analysis (Table 4) via a program called GOSummaries (trichdb.org).

**Figure 4 pathogens-12-00692-f004:**
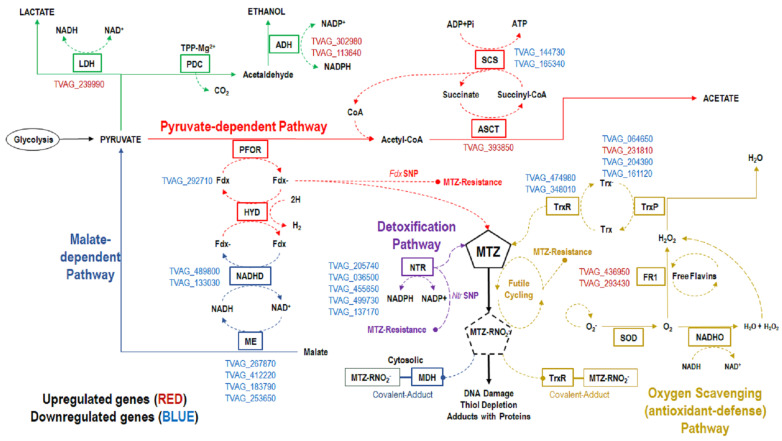
MTZ-resistance-related metabolic pathways in *T. vaginalis*. Resistance-related DEGs and their corresponding pathways connected to 5-nitroimidazole activation. Main energy production pathway—Pyruvate-dependent (Red), Malate-dependent (Blue); alternative energy production pathways (Light Green); oxygen-scavenging (antioxidant/redox) pathway (Gold); detoxification pathway (Purple); activation of MTZ through reduction (electron donation) resulting in formation of nitro radical anion and subsequent DNA damage. Highlighted boxes/words—proteins/enzymes with known dysregulated expression. Abbreviations: DEGs = differentially expressed genes.

**Table 1 pathogens-12-00692-t001:** *T. vaginalis* isolates and susceptibility to 5-nitroimidazoles.

Isolate	MLC (µg/mL)			Susceptibility Status	Source
	^1^ MTZ	^2^ TDZ	^3^ SEC		
1021	0.8	1.6	3.1	Sensitive	UAB
1003	6.3	1.6	1.6	Sensitive	UAB
1012	6.3	1.6	3.1	Sensitive	UAB
904	12.5	0.4	1.6	Sensitive	CDC
4448	100	50	25	Resistant	UAB
1073	200	100	100	Resistant	UAB
4266	400	50	100	Resistant	UAB
252	400	12.5	25	Resistant	CDC

^1^ MTZ—MLCs ≥ 50 µg/mL = resistance; ^2^ TDZ—MLCs ≥ 6.3 µg/mL = resistance; ^3^ SEC—resistance breakpoint not officially determined; our ongoing study suggests that a SEC MLC > 25 μg/mL correlates with resistance (unpublished data). Abbreviations: MLC = minimum lethal concentration, MTZ = metronidazole; TDZ = tinidazole; SEC = secnidazole; UAB = University of Alabama at Birmingham; CDC = Centers for Disease Control and Prevention.

**Table 2 pathogens-12-00692-t002:** Top dysregulated genes in MTZ-R *T. vaginalis* isolates.

Gene ID		Gene Product	Log2FC	*p*-Value *
Upregulated Genes	*TVAG_185520*	conserved hypothetical protein	4.80	3.33 × 10^−9^
	*TVAG_174500*	conserved hypothetical protein	3.48	3.26 × 10^−8^
	*TVAG_003210*	conserved hypothetical protein (SANT/Myb Homeobox-like domains)	6.11	1.14 × 10^−7^
	*TVAG_303800*	conserved hypothetical protein	6.63	6.49 × 10^−7^
	*TVAG_064800*	conserved hypothetical protein (Galactose-binding-like domain)	5.67	5.66 × 10^−6^
	*TVAG_191000*	conserved hypothetical protein (EGF-like domain)	4.75	6.46 × 10^−6^
	*TVAG_474560*	leucine-rich repeat protein, BspA family	5.01	9.60 × 10^−6^
Downregulated Genes	*TVAG_345450*	50S ribosomal protein L14p, putative	−7.19	3.29 × 10^−13^
	*TVAG_474000*	30S ribosomal protein S4p, putative	−6.52	2.82 × 10^−7^
	*TVAG_604680*	conserved hypothetical protein (Shisa-like protein)	−5.24	2.12 × 10^−6^
	*TVAG_345440*	50S ribosomal protein L14, putative	−3.83	2.19 × 10^−6^
	*TVAG_054400*	conserved hypothetical protein	−5.18	2.19 × 10^−6^
	*TVAG_108140*	conserved hypothetical protein	−4.59	4.42 × 10^−6^
	*TVAG_070260*	conserved hypothetical protein (Galactose-binding-like domain)	−5.64	7.47 × 10^−6^

* Most significantly differentially expressed genes, −Log10(*p*-value) > 5.

**Table 3 pathogens-12-00692-t003:** Unique genes identified from metabolic pathway enrichment analysis of 134 upregulated genes in MTZ-R *T. vaginalis* isolates.

Pathway Name	Gene IDs	Fold Enrichment	*p*-Value
Thiamine metabolism	TVAG_162060, TVAG_222600	8.4	0.0216
Methane metabolism	TVAG_302980, TVAG_472380	7.08	0.0298
Biosynthesis of type II polyketide backbone	TVAG_302980	30.44	0.0324
Nitrotoluene degradation	TVAG_302980	29.52	0.0334
O-Antigen nucleotide sugar biosynthesis	TVAG_222600, TVAG_302980	6.52	0.0348
Primary bile acid biosynthesis	TVAG_302980	27.06	0.0364
Glycolysis/Gluconeogenesis	TVAG_302980, TVAG_472380	5.99	0.0406
Linoleic acid metabolism	TVAG_302980	20.72	0.0473

**Table 4 pathogens-12-00692-t004:** Unique *T. vaginalis* genes identified from metabolic pathway enrichment analysis of 170 downregulated genes.

Pathway Name	Gene IDs	Fold Enrichment	*p*-Value
Riboflavin metabolism	TVAG_036500, TVAG_072960, TVAG_205740	7.55	0.00686
Fructose and mannose metabolism	TVAG_063860, TVAG_067220, TVAG_217780, TVAG_284100, TVAG_379200	3.39	0.0133
Aminoacyl-tRNA biosynthesis	TVAG_024820, TVAG_040800, TVAG_100390, TVAG_494870, TVAG_497170	3.2	0.0167
Toluene degradation	TVAG_063860, TVAG_214810, TVAG_217780	4.82	0.0230
Terpenoid backbone biosynthesis	TVAG_063860, TVAG_100390, TVAG_217780, TVAG_528020	3.07	0.0377

**Table 5 pathogens-12-00692-t005:** Dysregulated MTZ-resistance-related genes.

Pathway	Gene ID	Gene Product	Log2FC	*p*-Value
Carbohydrate/ Energy Metabolism	*TVAG_292710*	Ferredoxin 4 (fdx)	−1.56	0.015
	*TVAG_489800*	NADH dehydrogenase 51 kDa subunit (nadhd)	−1.05	0.048
	*TVAG_133030*	NADH-ubiquinone oxidoreductase flavoprotein, putative (nadhd)	−0.99	0.018
	*TVAG_267870*	malic enzyme, putative (me)	−1.40	0.011
	*TVAG_412220*	malic enzyme, putative (me)	−1.07	0.016
	*TVAG_183790*	malic enzyme (AP65-3 adhesin) (me)	−1.20	0.036
	*TVAG_253650*	malate dehydrogenase, putative (me)	−0.77	0.038
	*TVAG_239990*	malate dehydrogenase, putative (ldh)	1.34	0.047
	*TVAG_302980*	alcohol dehydrogenase, putative (adh)	3.96	0.0002
	*TVAG_113640*	alcohol dehydrogenase, putative (adh)	1.58	0.028
	*TVAG_393850*	acetyl-CoA hydrolase, putative (asct)	1.51	0.028
	*TVAG_144730*	succinate thiokinase, beta subunit (scs)	−1.27	0.015
	*TVAG_165340*	succinate thiokinase a subunit (scs)	−1.08	0.015
Detoxification	*TVAG_205740*	conserved hypothetical protein (ntr)	−3.52	0.0001
	*TVAG_036500*	conserved hypothetical protein (ntr)	−4.17	0.0008
	*TVAG_455650*	conserved hypothetical protein (ntr)	−1.96	0.015
	*TVAG_499730*	nitroreductase family protein (ntr)	−1.60	0.032
	*TVAG_137170*	conserved hypothetical protein (ntr)	−1.04	0.043
Oxygen Scavenging (antioxidant/ redox)	*TVAG_436950*	conserved hypothetical protein (fr1)	1.14	0.040
	*TVAG_293430*	conserved hypothetical protein (fr1)	1.63	0.043
	*TVAG_064650*	conserved hypothetical protein (trx)	−1.48	0.0069
	*TVAG_231810*	protein disulfide isomerase, putative (trx)	1.54	0.026
	*TVAG_204390*	thioredoxin m(mitochondrial)-type, putative (trx)	−1.01	0.034
	*TVAG_161120*	conserved hypothetical protein (trx)	−1.08	0.047
	*TVAG_474980*	dihydrolipoamide dehydrogenase, putative (trxr)	−1.35	0.005
	*TVAG_348010*	disulfide oxidoreductase, putative (trxr)	−0.82	0.042

## Data Availability

Next-generation sequencing raw and curated data are shared through GEO, accession number GSE227448.

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
