# Peer review of "Identification of Trichomonas vaginalis 5-Nitroimidazole Resistance Targets"

_pathogens, 2023, doi:10.3390/pathogens12050692_

Round 1

Reviewer 1 Report

Identification of Trichomonas vaginalis 5-nitroimidazole Resistance  Targets

Comments

This work is an important study in which the authors tried identifying genes contributing to 5-nitroimidazole resistance in the urogenital parasite Trichomonas vaginalis. Using  RNA sequencing, the authors identified 304 DEGs, 134 upregulated genes, and 170 downregulated genes in the resistant isolates.

The authors claimed future studies with more T. vaginalis isolates with a broad range of MLCs would be needed to determine which genes may represent the best alternative targets in drug-resistant strains. 

Reviewer 2 Report

In this paper, Graves et al. conducted a comparative RNAseq study with four metronidazole-sensitive and four metronidazole-resistant clinical isolates. It is great that the Division of Infectious Diseases of the UAB and the CDC undertake this initiative because they have access to a very large number of T. vaginalis isolates – more than this reviewer can even dream of. The experimental design is straight-forward and the study was well executed. Thus, in principle, the publication of this study will be a useful addition to the field. That said, I would also like to express some reservations:

1. The authors concentrate on mRNA expression only. Unfortunately, this does not always directly translate into protein levels. In a study on flavin reductase in T. vaginalis (Leitsch et al., 2014), flavin reductase 1 activity was consistently down in resistant isolates, but mRNA levels were not decreased as compared to sensitive strains in three out of five resistant strains (see supplementary table 3!). As FR1 has been now confirmed as factor in clinical metronidazole resistance in several independent studies, this means that the present study likely misses a lot of potential resistance factors. I would urge the authors to use the same strains in a proteomic study and to compare overall protein expression in all strains. Until this has been done I would advise the authors to be very careful when drawing conclusions or preparing graphical summaries as shown in Figure 4.

2. Regarding Figure 4: if the authors insist on showing a summary like this before proteomic data are available, I would recommend to concentrate on factors of clinical/aerobic resistance only. As the authors rightly state in the manuscript, two different modes of metronidazole resistance exist in T. vaginalis: “aerobic” and “anaerobic”. The latter is, strictly spoken, of biochemical interest only because “anaerobically resistant” strains are not viable in the host. It is important to keep these two resistance modes apart from each other, especially in the clinician’s view. This is also important when discussing a possible role of ABC transporters (lines 218ff). Clinical resistant strains are susceptible in the presence of oxygen only which is irreconcilable with a role for drug export by ABC transporters whose activity is independent of oxygen concentrations.

3. The genes found to be differentially expressed should be discussed a bit more carefully. When conducting a BLAST search with the TVAG_302980 sequence, e.g., I found that it is designated as a dehydroquinoate synthase, an enzyme of the ADH family. It is important to understand that ADHases are a large and diverse group of enzymes and without any supporting data it is premature to conclude that a given ADH produces ethanol. The authors might read into this a bit more because they seem to unaware that the ADH described in reference 25 is a secondary alcohol dehydrogenase, termed ADH-1. ADH-1 is a zinc-dependent enzyme rather than an iron-dependent enzyme like TVAG_302980.

4. I was puzzled by the use of the word “significant” in the paper. In the volcano plot (Figure 1) only a very small number of proteins are interpreted as “significantly” up- or downregulated. These proteins are also listed in table 2. In table 5 suddenly a new set of “significantly dysregulated” genes is shown. I assume that the authors used the word “significant” in the sense of “strongly” in Figure 1 but in the sense of “statistically significant” in Table 5. Please, do explain!

Reviewer 3 Report

The manuscript describes the identification of targets related to 5-nitroimidazole resistance in Trichomonas vaginalis. The 5-ni troimidazole resistance is  increasingly being reported clinically and recognized in in-vitro studies even upto 10% of infections. Thus it is very urgent and important to study the drug resistance to this class of antimicrobial in addition to seraching for drugs with different mechanisms of action. 

The work described is relevant and explained well. However, the number of isolates used in the study are less and it would be better if the experimets  could be performed a larger number of isolates. 

Neverthe less the work is important and results are useful for the advancement of science

satisfactory english 

Reviewer 4 Report

In this study, the authors utilized a traditional approach of investigating various clinical isolates to examine the genetic factors associated with resistance to 5-nitroimidazole in Trichomonas vaginalis. They performed an RNAseq-based analysis to identify new targets for future research and were able to predict a specific pathway that may be responsible for 5-nitroimidazole resistance in this organism. The study found promising results in identifying potential targets for combating resistance in Trichomonas vaginalis.

Major concerns:

The in vitro concentrations used are 100, 200 or 400 ug/mL, without explanation as to the choice of different concentrations in different experiments and without any discussion as to whether the need for different doses may suggest a dissociation among the effects that the authors interpret as inter-related.

The clinical strains utilized in this study have not been confirmed through either genome sequencing or sequencing based on 16sRNA.

The fact that resistance strains and susceptible strains exhibit varying levels of certain genes at the transcriptional level, and that these levels may or may not be linked to phenotypic resistance, implies that CFUs based assay is necessary to confirm the resistance and susceptibility phenotype of the strains after treatment with 5-nitroimidazole.

A principal component analysis of RNA seq data can be included to exhibit clusters of resistance and susceptible clinical strains, showcasing the similarity or differences of samples before and after treatment with 5-nitroimidazole.

The authors have not elaborated on the measures taken to assess the quality and integrity of the RNA prior to proceeding with RNA sequencing.

To enhance the study, it would be beneficial to present the leading candidates from one upregulated and one downregulated knockdown or knockout strain, demonstrating phenotypic resistance. Additionally, validation of the findings through qRT-PCR under resistant conditions could also improve the study.

Minor concerns-

The display of non-significant differentially expressed genes (DEGs) in the Venn diagram alongside the volcano plot appears confusing and unsuitable. It would be more appropriate to only show the significant number of DEGs (i.e., 7 up and 7 down) in the Venn diagram.

The authors did not provide an explanation for why they did not proceed with further experiments involving tinidazole and secnidazole beyond the MLC-based assay.

To validate their RNA sequencing data, the authors could perform qRT-PCR on four susceptible and four resistant isolates, particularly considering that some of the strains exhibit varying concentrations of resistance to 5-nitroimidazoles.

The manuscript lacks an explanation for why the author exclusively utilized anaerobic-induced resistance strains, and the protocol used is not well-detailed.

The introduction section does not provide a description of how environmental factors such as pH and nutrient availability can affect the development of resistance in vivo.

It may be confusing to include information about tinidazole and secnidazole (table1) in the main section of the study since they were not used for RNA sequencing. It would be more appropriate to include this information in the supplementary table.

The resolution quality of Figure 1 is poor, and utilizing color-based annotation instead of black, grey, and light grey could enhance the quality of Figures 2,3 & S1.

The discussion section and future prospects of the study could be enhanced by providing a more detailed explanation of the potential translation possibilities of the findings and suggesting additional experiments to investigate the role of metabolites in resistance.

The overall quality of English in this manuscript is good.

Reviewer 5 Report

The manuscript entitled "Identification of Trichomonas vaginalis 5-nitroimidazole Resistance Targets" Title, abstract and overall rationale of work is well written. However, there are still some minor concerns, which needs to be addressed before publication

1) Introduction section: This section is written well however, author wrote to much and I suggest the author to reduce the introduction and write concise way.

2) In the material methods section: Author need to explain the quality of RNA how they check or they did not check. If they check through gel then provide the gel picture.

3) The material method section is short and not descriptive and many thing are missing. I suggest author to incorporate details about the how to performance sequencing, library preparation and running time etc.

4) Sample size is very low and I suggest author to incorporate more sample to show clear scenario or difference between Trichomonas vaginalis 5-nitroimidazole Resistance and sensitive.

5) Author need to provide ethical approval number because this is human study.

6) Results section: Figure 2 and 3 should be present better way.

7) Author need expend discussion and much more explanations and interpretations must be added for the results, which are not enough at all. It is suggested to compare the results of the present research with some similar studies which is done before.

8) Some references are too long and author need to revise for example reference no. 6, 18 and other. I suggest author to revise if other latest manuscript is available in the same information.

English is written well and understandable 

Round 2

Reviewer 4 Report

We appreciate your response to our previous review and for taking the necessary steps to address the raised concerns. Your dedication in revising the manuscript is highly valued, and we have conducted a thorough evaluation of the changes made.

Congratulations are in order for the revisions made, and we are confident that the paper is now fit for publication. We are impressed with the efforts you have put in and believe that your work will make a valuable contribution to the field.

Once again, we thank you for your hard work and wish you success in your research endeavours.

Reviewer 5 Report

Author improve the quality of the manuscript but still issue in the figure 2 and 3. I am not able to differentiate clearly which one is highly up-regulate and which one low up-regulated due to low contract differentiation. I recommend author, they should change colour combination or presentation like red and blue or something else to show clear picture.

English is ok
